# Cross-Sectional Survey of the Training Practices of Racing Greyhounds in New Zealand

**DOI:** 10.3390/ani10112032

**Published:** 2020-11-04

**Authors:** Anna L. Palmer, Chris W. Rogers, Kevin J. Stafford, Arnon Gal, Darryl J. Cochrane, Charlotte F. Bolwell

**Affiliations:** 1School of Veterinary Science, Massey University, Private Bag 11-222, Palmerston North 4442, New Zealand; c.w.rogers@massey.ac.nz (C.W.R.); c.bolwell@massey.ac.nz (C.F.B.); 2School of Agriculture and Environment, Massey University, Private Bag 11-222, Palmerston North 4442, New Zealand; k.j.stafford@massey.ac.nz; 3Department of Veterinary Clinical Medicine, College of Veterinary Medicine, University of Illinois at Urbana-Champaign, 1008 W Hazelwood Drive, IL 61802, USA; agal2@illinois.edu; 4School of Sport, Exercise and Nutrition, Massey University, Private Bag 11-222, Palmerston North 4442, New Zealand; d.cochrane@massey.ac.nz

**Keywords:** greyhound, racing greyhound, training

## Abstract

**Simple Summary:**

There is a limited amount of scientific literature about the training of racing greyhounds. Previous reports have focused on racing injuries, race-track designs, and genetic traits of racing greyhounds, with little attention to training practices. Training and racing workload have been suggested as important factors associated with racing greyhound welfare and success. In this study, training practices of racing greyhounds were described by New Zealand trainers using a pro forma survey. We found that trainers considered similar factors, (1) the ability to reach time milestones and (2) the appearance of young dogs, which indicated when they were ready to begin formal race training and racing, to be important when training young greyhounds. Training programmes for race-fit greyhounds were structured around a weekly cycle of two gallop workouts or races a few days apart, separated by walking and free exercise. Training practices appear to be specific to the metabolic and physiologic adaptations required for the challenges associated with sprint racing. This description of training practices provides baseline information about the workload of racing greyhounds in New Zealand.

**Abstract:**

The aim of this study was to conduct a cross-sectional survey of racing greyhound trainers in New Zealand in order to provide an overview of their training practices. A survey regarding training practices was posted to all registered greyhound training license holders in New Zealand in August 2019. Data were collected from a convenience sample of 48 trainers (35.6%; *n* = 48/137) who completed the survey. Other than the differences in the number of greyhounds in race training, the training programmes described by public trainers and owner trainers were similar. Trainers reported that the primary reason for registering young dogs for racing and for qualifying for racing was the ability to meet time milestones. Young dogs had a median of six (interquartile range (IQR): 4–10) trials before they commenced their racing career. Trainers described training practices that aimed to prepare greyhounds for race-day. Regardless of whether the dogs raced once or twice a week, most training programmes demonstrated high specificity where training involved two periods of load cycles through high-intensity workload. Trainers racing their greyhounds once a week simulated the workload of trainers racing their greyhounds twice a week by introducing one high-intensity (speed) workout during the week. Training programmes were structured to condition the dogs to the physiological and metabolic requirements of sprint racing. This study highlights the importance of the need for an improved understanding of training and competition load in order to enable future research in the field of racing greyhounds.

## 1. Introduction

Generations of selective breeding have accentuated the physical attributes that allow greyhounds to be elite sprint athletes [1,2]. Their instinctive drive to chase [3] combined with their ability to gallop at high speeds for a short duration has resulted in the sport of greyhound racing. Greyhounds are capable of reaching speeds of approximately 18 m per second [4] and achieve anaerobic metabolism during a race [5]. The greyhound racing industry in New Zealand is highly regulated, where the pattern of racing and opportunities to race remain consistent throughout the year [6]. In New Zealand, greyhounds are raced, on average, every seven days, over distances between 295 m and 779 m [6]. It has been suggested that training and racing frequency contribute to unfavourable dog welfare outcomes during races [6,7]. There is minimal information, however, describing how greyhounds are prepared before and between races to test this hypothesis. 

Exercise accumulated during training and racing has been demonstrated to influence racing performance [8,9,10] and the risks of musculoskeletal injury [11] in thoroughbred racehorses. A conditioning programme is designed with the primary aim of stimulating the specific physiological adaptations required for performance [12,13]. For optimising performance, training sessions need to be correctly designed with well-balanced rest periods [13]. An understanding of the interaction between training volume (training intensity and frequency) and recovery periods can help identify training practices that avoid adverse dog welfare outcomes and is required to enhance performance [13,14] in racing greyhounds. 

In New Zealand, the training and racing of greyhounds is regulated by the governing body for greyhound racing (Greyhound Racing New Zealand) and the Racing Integrity Unit. There are two types of trainer licenses in New Zealand, an owner trainer license and a public trainer license. Owner trainers are trainers that own at least a share in the greyhound or greyhounds in their training kennel. Public trainers are trainers who race and train greyhounds for themselves as well as members of the public. Trainers must abide by the Greyhound Racing New Zealand Rules of Racing and Welfare Standards, which specify the duty of care required to meet the physical, health, environmental, behavioural and mental needs of greyhounds under the jurisdiction of Greyhound Racing New Zealand [15]. Kesl [1] recognised that the training programmes of racing greyhounds vary widely, in part due to the dog’s athletic ability but also due to the convenience of the trainers. Previous reports have detailed exercise protocols used in research studies [1,16]; however, no detailed training information has been recorded at either trainer or dog level. Cross-sectional surveys have been used as a preliminary step to describe training practices in other animal sports, including thoroughbred horse racing [17], standardbred horse racing [18] and endurance horse competitions [19,20]. Given the sparse documentation in scientific literature regarding racing and training practices of greyhounds, describing such practices will help understand the workload of racing greyhounds and is important to enable future research in this topic. The aim of this study was to provide an overview of the training practices of racing greyhound trainers in New Zealand and to examine if there were differences in practices between trainers with a public training license and an owner trainer license.

## 2. Materials and Methods

A self-administered postal survey (available in Appendix A) was distributed to all greyhound trainers (*n* = 137) that held a training license for the 2019/2020 racing season (August–July) with Greyhound Racing New Zealand. To increase the response rate, further convenience sampling occurred at 4 North Island racetracks, where trainers were approached on race day and asked if they were willing to participate. Follow-up phone calls were made with some trainers when surveys were returned incomplete. The potential for social desirability bias was recognised and efforts to reduce such bias included ensuring the survey responses remained confidential and self-administration of the surveys where possible.

The survey consisted of a combination of 21 open, closed, and multiple-choice questions, which took participants approximately 15 min to complete. The survey was divided into 3 sections; the demographics of the trainer (age, gender, type of training license, number of greyhounds in training), training of young dogs before racing (age young dogs are broken in, milestones), and the structure of a typical training programme for greyhounds during the racing season. The questions referred to the greyhounds that were in training at the time of the survey. 

The trainers’ answers were recorded and returned on a pro forma recording sheet and later entered into a Microsoft Excel (Microsoft, Redmond, WA, USA) spreadsheet for data manipulation. The exposure variable of interest was the type of license held by the trainer (referred to as license type), which was categorised by industry standards as either a public trainer or an owner trainer. Using these definitions “public trainers” can commercially train greyhounds for other people, whereas “owner trainers” only train their own greyhounds. Responses for free text answers were categorised into groups using key themes [21]. Data on training practices were grouped into categorical variables for racing frequency, number of low-intensity training sessions (including walking, free exercise, trotting, and swimming) and number of high-intensity workouts (galloping, trialling, or racing). The term trial refers to a controlled gallop event held at a racetrack where dogs run individually or as a competitive event, providing an opportunity for dogs to practice on the track. A qualifying trial is run under race conditions to select greyhounds ready to begin racing or for inclusion in a selected race. Nonparametric data were summarised with median and interquartile range (IQR), and categorical and binary data were summarised as counts and percentages. Associations between the exposure and outcome variables were assessed using Fisher’s exact test or Pearson’s χ^2^ test. A Kruskal–Wallis test was used to determine differences in the number of greyhounds in training for training license types. Variables showing association with the outcome with *p*-values < 0.05 were identified as exposure variables with a tendency towards statistical significance. The denominator for each question may vary as some questions did not apply to all trainers.

Statistical analyses were conducted in Stata version 15 (StataCorp. LP, College Station, TX, USA). 

## 3. Results

### 3.1. Respondent Demographics

A total of 48 trainers (35.6%; *n* = 48/137) completed the survey, three surveys were returned with insufficient detail to include in the analysis, two surveys were returned to sender and 84 did not respond. Of the 48 surveys included in the analysis, 37 responses were returned by post and 11 were conducted in person. Nine of the 37 posted surveys were followed up with a phone call to complete the survey and/or clarify unclear responses. Most trainers who responded (70.8%; *n* = 34/48) were from the North Island, and responses were received from eight regions across New Zealand. In total, 66.7% (*n* = 32/48) of the respondents were male and most were aged between 51 and 70 years old (Figure 1). The study population represented 663 actively racing greyhounds, which accounted for approximately 32% of the racing greyhound population in New Zealand in 2019. 

Respondents had a median of 20 years training experience (IQR: 10–30 years) and just under half of the trainers held a public training license (45.8%; *n* = 22/48). Public trainers tended to have more greyhounds in training (median: 16 greyhounds; IQR: 10–28) than owner trainers (median: six greyhounds; IQR: 3–8) (*p* = 0.0004). 

### 3.2. Training Facilities

Over half of the trainers (72.9%; *n* = 35/48) had a straight track (a long, narrow fenced off area where greyhounds can be hand slipped or free galloped) to exercise their greyhounds, with two trainers having two separate straight tracks. The median distance of a straight track was 180 m (IQR: 180–300 m). Most straight tracks (89.2%; *n* = 34/37) were flat, with only four (10.8%) on a hill. Thirty-five trainers (72.9%) used an exercise paddock to train greyhounds, and paddocks had a median size of 0.4 hectares (IQR: 0.1–0.8 ha). Four trainers (8.3%) reported having a circular training track. Most trainers (81.3%; *n* = 39/48) used a local racetrack for training, with a total of 10 different tracks around the country being used. Seven of these tracks were at locations of official greyhound race meetings and three were horse racing or training tracks that were used by greyhound trainers solely for the purpose of training. The different facilities used for training are shown in Table 1. 

### 3.3. Training Practices

#### 3.3.1. Preparing Young Dogs for Racing

Seventy-seven percent (*n* = 37/48) of respondents trained young dogs in preparation for racing. Greyhounds began race training at a median age of 12 months (IQR: 11–14 months) and met the milestones of box training, speed work, hand slipping at the track and trialling at a median age of 13 months (IQR: 12–14 months), 14 months (IQR: 12–15 months), 14 months (IQR: 13–16), and 16 months (IQR: 14–16 months), respectively. Most trainers (75.7%; *n* = 28/37) reported giving young dogs a break during the breaking in process for a median of four weeks (IQR: 2–6) and this occurred either during the breaking in process (42.9%; *n* = 12), before their qualifying trial (39.3%; *n* = 11) or after their qualifying trial (17.9%; *n* = 5).

Thirty-three trainers (89.2%) reported trials being used during training for education purposes and 40.5% (*n* = 15) of trainers used them for improving greyhounds’ fitness (Table 1). Greyhounds typically completed six trials (IQR: 4–10) before they qualified for racing. The primary reasons for greyhounds being registered and qualified for racing are reported in Table 1. For many trainers, the primary milestone used to decide when a greyhound was ready to be registered or for a qualifying trial was the greyhound’s ability to meet a minimum time (speed). 

Twenty-five trainers (67.6%; *n* = 25/37) reported training greyhounds with standardised training programmes with minor adjustments for specific dogs, eight trainers (21.6%; *n* = 8/37) reported using standardised training programmes for all greyhounds and four trainers (10.8%; *n* = 4/37) reported following specifically tailored programmes for each greyhound (Table 1).

#### 3.3.2. Training Race-Fit Greyhounds

Over half of the trainers (66.7%; *n* = 32/48) reported that a typical greyhound will race once a week and 16 (33.3%; *n* = 16/48) trainers reported racing their greyhounds twice a week. Trainers exhibited a training micro-cycle structured around a weekly period (Figure 2). 

Regardless of racing frequency, most training programmes involved 2–3 periods of high-intensity workload (72.9%; *n* = 35/48) per week. Most trainers racing their greyhounds once a week introduced a high-intensity training session during the week (62.5%; *n* = 20/32) (Figure 2). Twenty-two trainers (45.8%) did not gallop greyhounds between racing starts, 12 of these were trainers that typically raced their greyhounds once a week and 10 were trainers that raced greyhounds twice a week. Twenty-nine (60.4%) trainers reported that they did not make changes to the training schedule in the 48 h leading up to a race.

There was no significant difference between trainers in the type of workout they gave their dogs before or after racing. Before and after a race, most training sessions involved a low-intensity workout (53.1%; *n* = 34/64 and 64.1%; *n* = 41/64, respectively), or the day off (43.8%; *n* = 28/64 and 31.3%; *n* = 20/64, respectively) (high-intensity workout: 3.1%; *n* = 2/64 and 4.7%; *n* = 3/64, respectively). The pattern of training in the days surrounding high-intensity workouts was similar to the pattern of training around race days, where most trainers gave greyhounds a low-intensity workout (82.6%; *n* = 29/35) or the day off (14.3%; *n* = 5/35) the day before a gallop, and a low-intensity workout (80.0%; *n* = 28/35) or the day off (14.3%; *n* = 5/35) the day after a gallop.

Greyhounds performed low-intensity workouts for a median of four days per week (IQR: 3–5), which included walking, trotting, free exercise in a paddock or straight track and swimming/hydrotherapy. The median distance of low-intensity workouts was 3000 m (IQR: 1800–4000 m). Walking included road walking, with a median of 3000 m (IQR: 1800–4000 m) covered per session, or treadmill, where the median was 15 min (IQR: 15–20 min) of time spent per session. 

Trainers used high-intensity workouts, including galloping, trials, and racing, a median of two times per week (IQR: 1–3). The frequency and distance of training activities stratified by the number of race events trainers competed their dogs in per week are shown in Table 2. 

## 4. Discussion

The aim of the study was to provide baseline data on the training practices of racing greyhound trainers in New Zealand. As far as the authors are aware, there are no previous studies detailing cross-sectional training information at either trainer or dog level, and the use of a survey provided the ability to collect data on training practices. Postal surveys of trainers can yield low response rates [22], so after the initial postal distribution of surveys, a convenience sample of trainers was used for this study. Conducting surveys at race meetings provided an efficient method for promoting the survey, an opportunity to increase participation and the ability to aid participants with any queries to reduce non-response and errors. Therefore, the sample population was not entirely random and may have resulted in some selection bias. However, the geographical distribution of trainers in this study is consistent with previous reports for greyhound trainers across New Zealand [23]. The distribution of trainers in the study reflects the regions where racetracks are located and where racing regularly occurs. At the time the study was conducted, 64.2% (*n* = 88/137) of trainers in the target population were based in the North Island of New Zealand, 61.4% (*n* = 94/153 trainers including both individuals from training partnerships) of trainers were male and 39.4% (*n* = 54/137) held a public training license. The gender of trainers included in this study and type of training license held were consistent with the wider population. The median trainer age group of the target population was 41–50 years, which is lower than reported in this study; however, the proportion of trainers across the different age groups was similar. The distribution of the sample population reflected the demographics of the target population of registered trainers and can therefore be considered representative of the racing greyhound trainer population in New Zealand. 

Training practices were mostly homogeneous with no significant differences between public trainers and owner trainers with regard to most of the facilities used for training and the general structure of the training programme. Such uniformity of training programmes has previously been described in racing standardbred horses [24] and thoroughbred horses [25]. There is potential that a social desirable bias may have influenced the results of this study, particularly where anonymity was lost when surveys were completed in person or through phone communication. Such bias would mean that trainers answered questions according to how they perceived society (or the interviewer) to want them, rather than representing the facts [26]. Social desirable bias cannot be ruled out as the reason why no differences were seen between training practices of public trainers and owner trainers. In our study, although the overall structure of training was reported, numerous trainers noted that the programmes can vary between individual greyhounds as well as greyhounds racing different distances (i.e., sprinting greyhounds compared with long distance (staying) greyhounds). Despite overall training programmes being similar amongst the two groups of trainers, potential differences in the structure of training reported at individual dog level may occur.

Greyhounds have been selectively bred for speed and performance traits required for sprinting [2]. Similar to reports from thoroughbred racehorses [27,28,29], greyhounds appear to require only a few gallop-load cycles to stimulate the appropriate musculoskeletal responses to race training. The training practices for young dogs described in this study reflect the requirements of performance bred canine athletes. The introduction of gallop work and trialling provides an initial loading phase, which conditions the greyhounds and provides the skill and specificity required for their racing career within a limited number of load cycles. Our study found that during the conditioning phase of training, greyhounds typically commence training at 12 months of age and speed work at 14 months of age, which allows preparation for racing, which begins, according to industry standards, at 16 months of age (rule 19.10; [30]). Furthermore, greyhounds will partake in a median of six trials before they complete their qualifying trial and commence racing. In agreement with previous reports of athletic training [31], trainers noted that there was considerable variation between individual dogs, especially when training young dogs, in regard to the volume of training required to prepare the greyhound for racing.

Once the gallop and trial work commence, greyhounds maintain a regular pattern where load cycles endured during racing, trialling or galloping work are balanced by rest days or days where low-intensity work occurs (Figure 2). A recent study reported that greyhounds race every seven days [6], and thus once greyhounds reach a “maintenance” stage, trainers demonstrate a pattern where races and high-intensity workout sessions are used as a method to condition greyhounds, maintain fitness and provide load cycles. During high-intensity training sessions, greyhounds typically endure 70–91 load cycles (given that a greyhound’s stride length is 5 m and the median distance covered during high-intensity workouts is approximately 350–457 m). Greyhounds undertake 2 high-intensity workouts per week, either racing once and completing one gallop, or racing twice per week. Furthermore, despite most trainers reporting that changes were not made to the training programme in the days before a race, most trainers demonstrated tapering (reducing the intensity or no exercise) in the 48 h before a race. This difference could be explained because the day-off or reduced workload before a race was not considered a change in the weekly micro-cycle; rather, tapering was a routine practice demonstrated when training greyhounds. The racing schedule dictates this micro-cycle of high-intensity and low-intensity training sessions (Figure 2). 

Training programmes reported in this study appeared to be designed to improve anaerobic performance. Periods of high-intensity exercise provide opportunity for physiological adaptations required to enhance exercise performance. High-intensity workouts induce ATP (Adenosine Triphosphate) regeneration through the phosphagen and glycolytic systems [1,32,33], the activity of glycolytic regulatory enzymes, the accumulation of muscle and blood lactate [1,34], and changes in muscle pH [1]. The primary method of energy production for high-intensity performance is glycolytic metabolism [35]. Greyhound limbs contain a large proportion of fast-twitch type IIA muscle fibres [36,37] and such muscle fibres have notable glycolytic capacity [5]. The reported training programmes were tailored to short duration bouts of anaerobic metabolism, which imitates the physiological requirements during racing, and we can therefore conclude that training programmes are specific to the metabolic system (Figure 3). 

Furthermore, the training programmes appeared to be designed to prevent muscular injury, given that the risk of injury increases with increasing load cycles [7,39] (Figure 4). Greyhounds are exposed to a high number of cumulative load cycles during training and racing. The magnitude and nature of these loads are influenced by the accumulation of exercise, training volume, rest periods and environmental factors [13,31,40]. The training programmes reported here were consistent with practices to reduce muscular injury. Previous studies have reported that most of the injuries recognised on race-day by veterinarians are categorised as general soreness affecting soft tissue and could be considered a by-product of athletic pursuit [41,42]. The ratio of high-intensity workouts to low-intensity workouts suggests that the low-intensity sessions are used as an active recovery phase. Activity, commonly walking, is completed to allow a range of movements, to mitigate delayed onset of muscle soreness, to reduce the occurrence of bruising and to increase blood flow, which helps to flush out chemical waste [43].

It is likely that trainers use races to achieve specificity and stimulate appropriate physiological adaptations to maintain or improve performance. Our study found that during the performance phase, where greyhounds are racing, they take part in repeated micro-cycles of high-intensity workouts including gallop sessions and racing. The structure of the micro-cycle was dictated by the opportunities to race. Races were used as a method to condition greyhounds and periods of low-intensity work that occurs between the load cycles are an active form of recovery where the trainers are managing their canine athletes. Previous work has demonstrated that, regardless of the age the greyhound began racing, greyhounds of a similar ability finished racing at a similar age [6]. Similar to all athletes, greyhounds have a period where they are at their physiological prime for the required speed work and the length of a racing career in greyhounds is limited by the age of the greyhound [6,44]. Regular racing during the window of time where greyhounds are at peak performance offers the greatest possible economic return for the trainer and/or owners of the greyhound. However, regular racing during this period leads to the accumulation of load cycles and thus an increasing risk of injury. The convergence at which the risk of injury increases due to accumulating load cycles is unknown for racing greyhounds and is an area where future work should be focused. Further prospective data capture of training regimes and physiological markers is required to examine the relationship between the quantity of load cycles, training volume and risk of injury.

## 5. Conclusions

This study provided baseline data on the training practices of racing greyhound trainers and highlighted factors unique to the greyhound racing industry in New Zealand. There were no significant differences between the training practices of trainers holding a public training license compared with owner trainer license holders. The training structure of racing greyhounds appears to be influenced by race-days, which help create a weekly micro-cycle. Regardless of whether a dog is racing once or twice a week, most training programmes demonstrated high specificity where training involved two high-intensity workload periods of load cycles. Periods of high-intensity exercise during training regimes are designed to improve anaerobic performance and provide opportunity for physiological adaptations required to enhance exercise performance. This study provides the necessary baseline data for future studies to explore. 

## Figures and Tables

**Figure 1 animals-10-02032-f001:**
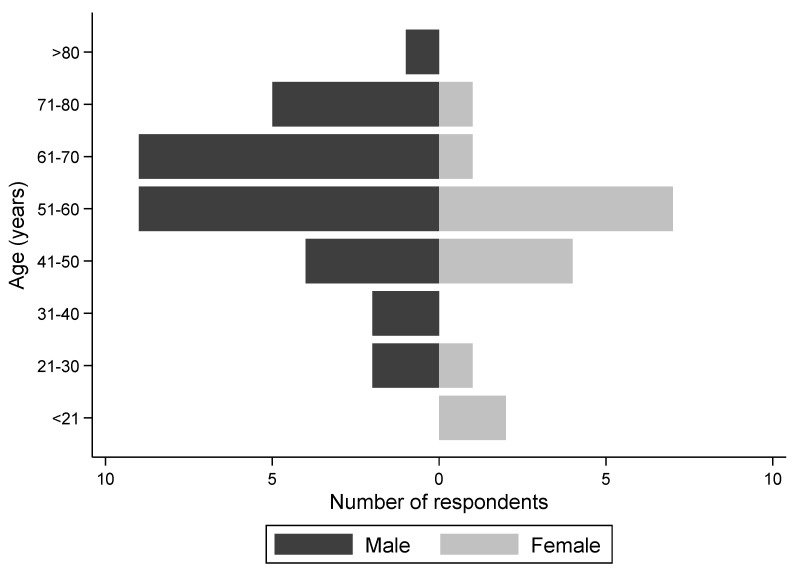
Population pyramid for age and gender of the 48 trainers from a cross-sectional survey of the training practices of racing greyhounds in New Zealand.

**Figure 2 animals-10-02032-f002:**
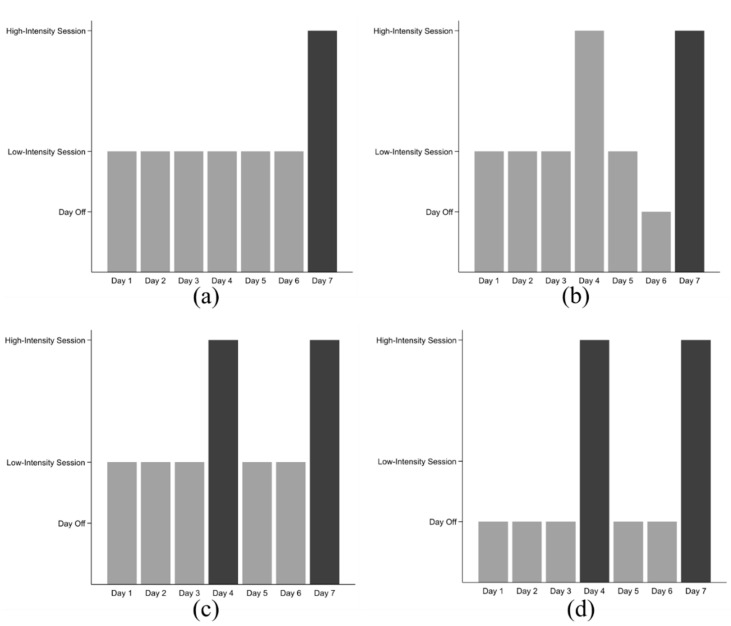
The four primary models of the training micro-cycle as described by 48 greyhound trainers across New Zealand. Racing days are highlighted by the dark grey bars and training days are shaded light grey; (**a**) low-intensity based training programme with 1 race (day 7) per week; (**b**) trainers give greyhounds 1 mid-week high-intensity training session; (**c**) low-intensity based training system with 2 race days per week; (**d**) training system where greyhounds race twice each week and do not participate in any workouts between race-days; (**e**) represents a typical training micro-cycle. The high-intensity training or racing load is indicated by the peaks, which are followed by a plateau of low-intensity work, which allows a period for rehabilitation and recovery before the next training load. Two of these cycles are typically followed by a rest period before the next load cycle. The racing schedule dictates the micro-cycle where greyhounds have 2 rest or low-intensity days to 1 high-intensity day.

**Figure 3 animals-10-02032-f003:**
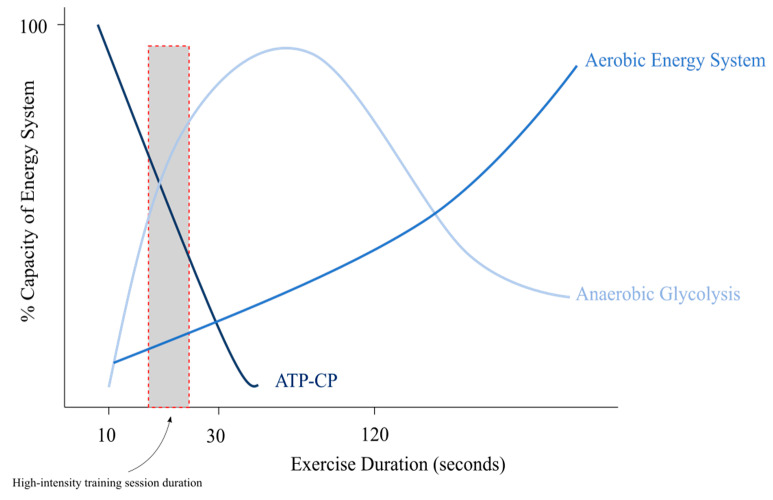
Training of the metabolic systems involved with energy production. During high-intensity workouts, the threshold at which anaerobic glycolysis becomes the predominant system for energy production is reached. Figure adapted from [38].

**Figure 4 animals-10-02032-f004:**
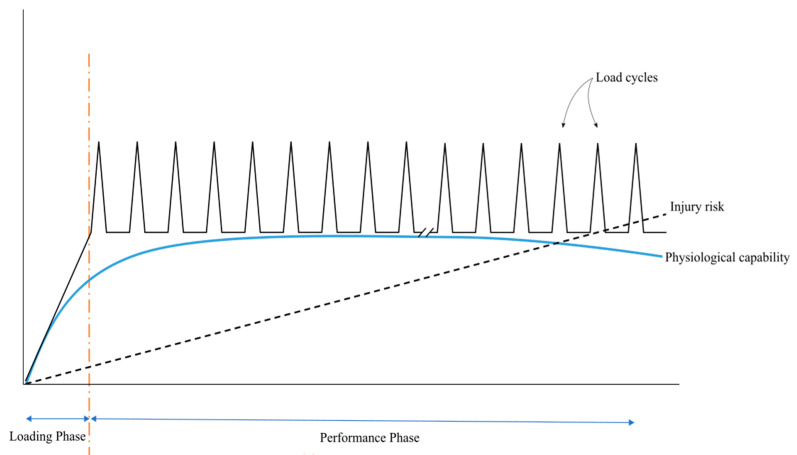
Loading and performance stages of greyhound training and racing. Young dogs are introduced to the stressors of racing during the load phase, and after 6–7 load cycles they commence racing in the performance phase. While the load cycles create a performance curve that is limited by the greyhound’s physiological capabilities, with increasing number of load cycles and little time for recovery, the risk of injury increases.

**Table 1 animals-10-02032-t001:** Descriptive demographics, facilities used, and training methods of racing greyhounds as reported by 48 trainers from a cross-sectional survey of the training practices of racing greyhounds in New Zealand. Pearson’s χ^2^ or Fisher’s exact test *p*-values are reported to demonstrate differences between the two groups of licensed trainers.

Variable	Total	Public Trainer	Owner Trainer	*p*-Value
Number of Trainers	48	22 (45.8)	26 (54.2)	
Location of Trainers				0.227
Auckland	5 (10.4)	4 (8.3)	1 (2.1)	
Waikato	14 (29.2)	8 (16.7)	6 (12.5)	
Bay of Plenty	1 (2.1)	0	1 (2.1)	
Taranaki	3 (6.3)	0	3 (6.3)	
Manawatu-Whanganui	11 (22.9)	4 (8.3)	7 (14.6)	
Canterbury	11 (22.9)	5 (10.4)	6 (12.5)	
Otago	2 (4.2)	0	2 (4.2)	
Southland	1 (2.1)	1 (2.1)	0	
Facilities used for training				
Straight Track				0.978
Yes	35 (72.9)	16 (33.3)	19 (39.6)	
No	13 (27.1)	6 (12.5)	7 (14.6)	
Exercise Paddock				0.049
Yes	35 (72.9)	13 (27.2)	22 (45.8)	
No	13 (27.1)	9 (18.8)	4 (8.3)	
Training Track				0.371
Yes	4 (8.3)	1 (2.1)	3 (6.3)	
No	44 (91.7)	21 (43.8)	23 (47.9)	
Treadmill				0.091
Yes	17 (35.4)	5 (10.4)	12 (25.0)	
No	31 (64.6)	17 (35.4)	14 (29.2)	
Beach				0.597
Yes	7 (14.6)	3 (6.3)	4 (8.3)	
No	41 (85.4)	19 (39.6)	22 (45.8)	
Local Track				0.113
Yes	39 (81.3)	20 (41.7)	19 (39.6)	
No	9 (18.8)	2 (4.2)	7 (14.6)	
Train young dogs				0.357
Yes	37 (77.1)	18 (37.5)	19 (39.6)	
No	11 (22.9)	4 (8.3)	7 (14.6)	
Factors that influence trainers’ decision to register young dogs for racing				0.347
Appearance	5 (10.4)	4 (8.3)	1 (2.1)	
Time milestones	17 (35.4)	8 (16.7)	9 (18.8)	
Weeks in training	2 (4.2)	1 (2.1)	1 (2.1)	
Owner decision	2 (4.2)	2 (4.2)	0	
All greyhounds are registered	9 (18.6)	3 (6.3)	6 (12.5)	
Age of the greyhound	1 (2.1)	0	1 (2.1)	
Factors that influence trainers’ decision to qualify young dogs for racing				0.722
Appearance	7 (14.6)	3 (6.3)	4 (8.3)	
Time milestones	23 (47.9)	11 (22.9)	12 (25.0)	
Weeks in training	1 (2.1)	0	1 (2.1)	
Owner decision	1 (2.1)	1 (2.1)	0	
All greyhounds are qualified	2 (4.2)	1 (2.1)	1 (2.1)	
Training regime before first race				0.61
Standardised	8 (16.7)	4 (8.3)	4 (8.3)	
Similar	25 (52.1)	11 (22.9)	14 (29.2)	
Personalised	4 (8.3)	3 (6.3)	1 (2.1)	
Difference in training programme for greyhounds running different distances				0.626
Yes	25 (52.1)	12 (25.0)	13 (27.1)	
No	22 (45.8)	9 (18.8)	13 (27.1)	
Difference in training programme within 48 h before a race				0.074
Yes	18 (37.5)	11 (22.9)	7 (14.6)	
No	29 (60.4)	10 (20.8)	19 (39.6)	
Training sessions recorded				0.446
Yes	19 (39.6)	7 (14.6)	12 (25.0)	
No	27 (56.3)	13 (27.1)	14 (29.2)	

**Table 2 animals-10-02032-t002:** Frequency and distance of training activities (excluding racing) of racing greyhounds as reported by 48 trainers from a cross-sectional survey of the training practices of racing greyhounds in New Zealand.

Training Activities	Racing Once a Week	Racing Twice a Week
Low-intensity training		
Median times per week	4 (3–5)	4 (2.25–5)
Median Distance (m) per training session	3000 (1875–4000)	3000 (1800–3250)
Total weekly distance (m)	7000 (2750–16,000)	9350 (6675–13,750)
High-intensity training		
Median times per week	2 (1–2)	2 (2–3)
Median Distance (m) per training session ^1^	457 (350–457)	457 (440–457)
Total weekly distance (m)	727 (457–907)	989 (914–1227.75)

^1^ Distance including race distances. Where race distances were not given in the survey response, the median race distance for greyhound races in New Zealand from Palmer et al. [6] was used (457 m). Seventy-seven percent (*n* = 37/48) of trainers reduced the workout intensity or gave greyhounds the day off in the 48 h before a race according to the weekly training schedules.

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
