# Peer review of "Cross-Sectional Survey of the Training Practices of Racing Greyhounds in New Zealand"

_animals, 2020, doi:10.3390/ani10112032_

Round 1

Reviewer 1 Report

The paper can be almost accepted for publication as it is. I have just 2 concerns/comments:

  1. I cannot find in your conclusions the aim stated in line 75: differences in practices between trainers with a public training license and an owner trainer license. Moreover, I would suggest clarifying the process/requirements to become a Public Trainer or Owner/Trainer of Greyhound Racing New Zealand (GRNZ). Are there significant differences?
  2. I would strongly recommend adding a section on dog welfare issues in the Greyhound industry, not only in terms of injuries but also in terms of behavioral needs. For this purpose, the authors should add some references and maybe answer the following questions:
    1. Are the regulations in New Zealand looking at greyhound racing welfare? (see https://assets.publishing.service.gov.uk/government/uploads/system/uploads/attachment_data/file/552984/greyhound-racing-consult-sum-resp.pdf)
    2. Are the GRNZ’s health and welfare standards adequate and satisfied?
    3. Are the GRNZ’s socialization and rehoming guidelines implemented?

Reviewer 2 Report

Review: Cross-Sectional Survey of the Training Practices of 2 Racing Greyhounds in New Zealand (Palmer et al.) 

This study surveyed racing greyhound trainers in New Zealand, with an aim to produce baseline data about training practicesA survey, consisting of 21 questions about respondent demographics, young dog training and training programme structure, was mailed to all New Zealand training licence holders, with additional recruitment at 4 tracks on race days. Responses were received from 48 trainers (35.6%). Descriptive results are presented for training location characteristics, deciding factors for registering and qualifying young dogs, and training schedules surrounding race day and low- and high-intensity sessions. The authors noted few differences between public and owner trainers. Results will form basis for future studies. 

Overall, this is a clear and well written paper. Although the goal was to produce baseline data, the discussion is impressively thorough and clearly places the results in context in terms of physiology; this section is particularly well done and I enjoyed reading it. 

I have only a few minor comments for you to address: 

  • Line 91: Did any respondents return incomplete surveys? If so, were they included in analysis? 
  • Line 113: How many respondents were a result of the postal survey and how many were recruited from the tracks? 
  • Line 113/208: How do other respondent demographics compare to demographics of all New Zealand trainers? Are they, and thus your results, representative of the population? For example, are the majority of NZ trainers male and between the ages of 51 and 70? 
  • Discussion: Could social desirability (providing the ‘right’ responses) have played a role in the responses you received? This may be especially true if respondents from in-person recruitment completed the survey in your presence; the procedure surrounding this should be added to the methods section for clarity 
  • Discussion: Have similar studies been conducted in other countries? I suspect the topic is under-researched, so I understand if not, but if so, a comparison to practices in other countries would be interesting given the international audience of the journal. 
